# Unseen Weapons: Bacterial Extracellular Vesicles and the Spread of Antibiotic Resistance in Aquatic Environments

**DOI:** 10.3390/ijms25063080

**Published:** 2024-03-07

**Authors:** Muttiah Barathan, Sook-Luan Ng, Yogeswaran Lokanathan, Min Hwei Ng, Jia Xian Law

**Affiliations:** 1Department of Medical Microbiology, Faculty of Medicine, University of Malaya, Lembah Pantai, Kuala Lumpur 50603, Malaysia; barathanmuttiah@um.edu.my; 2Centre for Tissue Engineering and Regenerative Medicine, Faculty of Medicine, Universiti Kebangsaan Malaysia, Cheras, Kuala Lumpur 56000, Malaysia; lyoges@ppukm.ukm.edu.my (Y.L.); angela@ukm.edu.my (M.H.N.); 3Department of Craniofacial Diagnostics and Biosciences, Faculty of Dentistry, Universiti Kebangsaan Malaysia, Jalan Raja Muda Abdul Aziz, Kuala Lumpur 50300, Malaysia

**Keywords:** antibiotic resistance (ABR), aquatic, bacterial extracellular vesicles (BEVs), microbial ecology, resistance genes

## Abstract

This paper sheds light on the alarming issue of antibiotic resistance (ABR) in aquatic environments, exploring its detrimental effects on ecosystems and public health. It examines the multifaceted role of antibiotic use in aquaculture, agricultural runoff, and industrial waste in fostering the development and dissemination of resistant bacteria. The intricate interplay between various environmental factors, horizontal gene transfer, and bacterial extracellular vesicles (BEVs) in accelerating the spread of ABR is comprehensively discussed. Various BEVs carrying resistance genes like *blaCTX-M*, *tetA*, *floR*, and *sul/I*, as well as their contribution to the dominance of multidrug-resistant bacteria, are highlighted. The potential of BEVs as both a threat and a tool in combating ABR is explored, with promising strategies like targeted antimicrobial delivery systems and probiotic-derived EVs holding significant promise. This paper underscores the urgency of understanding the intricate interplay between BEVs and ABR in aquatic environments. By unraveling these unseen weapons, we pave the way for developing effective strategies to mitigate the spread of ABR, advocating for a multidisciplinary approach that includes stringent regulations, enhanced wastewater treatment, and the adoption of sustainable practices in aquaculture.

## 1. Importance and Multifaceted Nature of the Aquatic Environment

The aquatic environment constitutes any natural or artificial setting predominantly comprising water, encompassing oceans, seas, lakes, rivers, streams, wetlands, and subterranean water reservoirs. This environment encompasses both freshwater and marine ecosystems [1]. Covering around 71% of the Earth’s surface, water consists largely of saltwater in the oceans, accounting for approximately 97.5%. Conversely, only about 2.5% of Earth’s water is freshwater, primarily distributed in rivers, lakes, and underground reservoirs. A minor fraction of this freshwater is readily accessible and suitable for human use [2]. Despite this limited accessibility, the aquatic environment constitutes a substantial portion of the Earth’s surface, playing a pivotal role in the planet’s dynamics. Supporting a diverse array of species, ranging from microorganisms like phytoplankton and zooplankton to larger organisms such as fish, amphibians, reptiles, birds, and mammals, many exclusive to aquatic ecosystems, it is estimated that over 50% of all known species inhabit aquatic environments [3]. These species contribute significantly to ecosystem equilibrium, offer sustenance, and enhance biodiversity. Additionally, aquatic ecosystems perform crucial functions like climate regulation through carbon dioxide absorption and oxygen release, water purification by filtering pollutants, the provision of habitats for commercially valuable fish species, and the facilitation of recreational activities [4]. The aquatic environment experiences influence from diverse factors including water temperature, salinity, pH, nutrient concentrations, currents, and anthropogenic activities. Safeguarding and conserving the aquatic environment are imperative to ensuring the resilience of ecosystems and the welfare of both aquatic life and human communities [5].

## 2. Antibiotic Resistance (ABR) in Aquatic Environments

Aquatic animals contribute to 17% of global animal protein consumption, with fish providing nearly 20% of per capita animal protein for over 40% of the world’s population. Global food fish consumption growth has outpaced the consumption of meat from terrestrial animal production sectors, except for poultry [6]. Since 2001, global aquaculture has grown at a rate of 5.8% annually, driven by increased demand for animal protein in fast-growing economies. Asia is the major contributor, accounting for nearly 90% of global aquaculture production, with China alone contributing 61% in 2016 [7].

The rising demand for animal source nutrition has led to increasing intensification in animal production systems, including aquaculture. The transitional period toward intensive production often involves the use of non-therapeutic antimicrobials to enhance growth and compensate for insufficient biosecurity and management practices. Antimicrobial use in terrestrial food-producing animals is already high and is expected to increase significantly by 2030, especially in fast-growing economies like BRICS countries (Brazil, Russia, India, China, and South Africa) [8]. 

Antibiotic use in aquaculture serves multiple purposes, including disease prevention, growth promotion, and water quality management. Disease prevention and treatment are vital in densely stocked aquaculture systems, where antibiotics are used to control bacterial infections [9]. Prophylactic antibiotic use helps mitigate known risks during activities like transportation or stocking. Additionally, subtherapeutic antibiotic doses can promote growth; however, these practices exert strong selective pressure, favoring the emergence and selection of antibiotic resistance (ABR) strains, with subsequent dissemination of ABR traits through different routes, such as food, feed, and the environment [10]. 

Antibiotics such as macrolides have 12–16 lactone rings and are lipophilic with low water solubility. They are generally bacteriostatic and are used to treat respiratory tract, skin, and soft tissue infections [11]. Beta-lactam antibiotics contain at least one beta-lactam ring. They interfere with cell wall synthesis and are active against a wide range of bacteria, making them widely used, with penicillin being a prominent member. In addition, sulfonamides, derived from a p-amino-benzene–sulfonamide functional group, are widely used, have a long decomposition half-life, and are commonly used in veterinary medicine [12]. Tetracyclines are amphoteric and unstable in bases but stable in acids. They are extensively used in veterinary and human medicine, in agriculture, and as food additives. They pose environmental challenges and may not be fully removed by wastewater treatment plants. Quinolones are fat-soluble and resistant to various environmental conditions. They are used for treating infectious diseases and promoting livestock and aquaculture, entering aquatic environments through wastewater and direct discharge [13].

China is the world’s largest animal-feeding country, with expanding and intensifying operations. Hence, a significant number of antibiotics, approximately 53,800 tons out of the 92,700 tons used, are released into the environment, including surface waters, groundwater, and coastal waters [14]. The most commonly detected antibiotics are tetracyclines, sulfonamides, and fluoroquinolones. Concentrations of antibiotics are generally higher in northern and eastern China, where the population density is higher. The main sources of antibiotics in water have been found to be wastewater from aquaculture and animal husbandry [15]. The antibiotic concentration in the Guangzhou area of the Pearl River is 3–4 times that of rivers in Europe and America [16]. A systematic review has mentioned that ABR isolates achieved nearly 92% of resistance towards antibiotics were detected in the region of China and have exceeded safe limits (Predicted No Effect Concentrations) in wastewater, treatment plant influents/effluents, and receiving environments. Wastewater and plant influent/effluent treatment plants have shown the greatest potential for resistance development. The tap and drinking water in the WPR and China have shown the highest levels and likelihood of exceeding PNECs, particularly for ciprofloxacin [17].

On the other hand, macrolides, sulfonamides, and trimethoprim have been frequently detected in the water of the Tama River in the Tokyo metropolitan area with concentrations ranging from 4 to 448 ng/L. Antibiotic residues are generally higher in urban rivers compared with rural ones. The dominance of human antibiotics from sewage effluents has been noted to be a contributing factor to the antibiotic composition in urban rivers, rather than veterinary medicines from livestock wastewater [18]. About 25% of wastewater treatment systems in Japan are combined sewer systems that carry both untreated wastewater and rainwater simultaneously. Combined sewer systems pose a high risk of causing environmental contamination, especially during heavy rains and flooding events, which have become more pronounced because of climate change [19]. However, there are no regulations limiting the discharge of antimicrobial resistance (AMR), and the actual amount of AMR is unknown, highlighting a potential gap in environmental monitoring and regulation.

AMR affects all countries, but the burden is higher in lower–middle-income countries (LMIC) such as Guatemala, Honduras, El Salvador, Panamá, Paraguay, Haiti, the Dominican Republic, Belize, Suriname, Uruguay, French Guiana, and Guyana because of various challenges. Studies have also mentioned that only 50–60% of Latin America is connected to sewage, and only 30% of domestic sewage is treated. A lack of financial resources in most countries poses challenges to the effective management of wastewater since it is one of the hot spots for AMR [20]. 

Indian rivers contain antibiotic residues that may contribute to the growing problem of AMR and may have negative effects on the ecosystem and human health [21,22]. Untreated pharmaceutical wastewater and hospital effluents are a major source of contamination. One study mentioned that three antibiotics, ofloxacin, norfloxacin, and sulfamethoxazole, were detected in river water samples from four Indian rivers. They were detected and found to be two to five times higher than the safe limit. 

Antibiotics and non-steroidal anti-inflammatory drugs (NSAIDs) are highlighted as the most frequently reported pharmaceuticals in African waters. Examples of detected antibiotics include sulfamethoxazole, with concentrations reaching 53.8–56.6 μg/L in Kenya and Mozambique. Amoxicillin, another antibiotic, has shown concentrations ranging from 0.087 to 272.2 μg/L in Nigeria. The NSAID ibuprofen was detected at concentrations of up to 67.9 and 58.7 μg/L in Durban City and the Msunduzi River (KwaZulu-Natal, South Africa), respectively. An antiretroviral drug, lamivudine, has reached concentrations of up to 167 μg/L in surface water samples from Nairobi and Kisumu City, Kenya. In Asian countries, antibiotics have been detected at concentrations reaching 365.05 μg/L in surface water samples. Concentrations of other pharmaceuticals in Asian environmental waters are generally lower compared with those in African waters [23].

Antibiotic usage has increased globally, and if no policy reforms are made, it is expected to reach 126 billion defined daily doses in 2030. Overall antibiotic pollution is shown in Figure 1. A significant portion (30–90%) of antibiotics is released into the environment, posing a threat to ecosystems. Antibiotics have been found in various environmental compartments, with water being the most commonly reported. Examples of frequently detected antibiotics include fluoroquinolones, tetracyclines, sulfonamides, macrolides, and beta-lactams [24]. Despite the substantial use of antibiotics in aquaculture, only a relatively small percentage (20~30%) is absorbed by the aquaculture products themselves. This low utilization rate suggests that a significant portion of antibiotics may not serve their intended purpose within the target organisms [25]. Hence, the use of antibiotics in both terrestrial and aquatic animal production is contributing to ABR, which is a major global health challenge. The presence of antibiotics in different environments depends on their physicochemical properties, including the octanol/water dividing coefficient (Kow), the distribution coefficient (Kd), separation constants (pKa), vapor pressure, and Henry’s law constant (KH). Stability and decomposition rates vary among different antibiotics [26]. For example, penicillin is easily decomposed, while fluoroquinolones and tetracyclines are more stable, leading to longer persistence and potential accumulation in the environment. Beta-lactam antibiotics, like penicillin, have beta-lactam rings that contribute to their degradation in the environment. In contrast, ciprofloxacin and erythromycin are more resistant to degradation because of the absence of beta-lactam in their structures. Fluoroquinolones and sulfonamides are highlighted as potentially dangerous antibiotics in the environment, but they may undergo degradation when exposed to sunlight [27,28].

In aquaculture, antibiotic use is driven by production intensification and the increasing incidence of aquatic animal pathogens. Prolonged antibiotic use in aquaculture has led to ABR among humans [29,30] since many antibiotics used in aquaculture, such as tetracycline, macrolides, and aminoglycosides, are critically important for human treatment according to the World Health Organization (WHO). The 2022 Global Antimicrobial Resistance and Use Surveillance System (GLASS) report reveals alarming resistance rates among prevalent bacterial pathogens worldwide. Infections from resistant bacteria kill 700,000 people every year, with over 90% of them in low- and middle-income countries. Key findings include a median reported rate of 42% for third-generation cephalosporin-resistant *Escherichia coli* (*E. coli*) and 35% for methicillin-resistant *Staphylococcus aureus* (MRSA) in 76 countries [31,32]. Urinary tract infections caused by *E. coli* show reduced susceptibility to standard antibiotics, posing challenges in treatment. *Klebsiella pneumoniae* (*K. pneumoniae*) exhibits elevated resistance levels against critical antibiotics, potentially leading to the increased use of last-resort drugs like carbapenems. Projections indicate a twofold surge in resistance to last-resort antibiotics by 2035. The urgent need for robust antimicrobial stewardship practices and enhanced global surveillance is underscored to address the growing threat of antibiotic resistance. Additionally, by 2050, ABR will be accountable for 10 million deaths annually and harm the economy in a manner similar to that of the global financial crisis of 2008–2009 [33].

Meanwhile, ABR in Southeast Asia (SEA) aquaculture involves 17 different drug classes, with the most commonly reported resistances occurring in aminoglycosides, beta-lactams, (fluoro)quinolones, tetracycline, sulfa groups, and multi-drug resistances. *E. coli*, *Aeromonas*, and *Vibrio* spp. are the most commonly reported bacteria resistant to antibiotics in SEA aquaculture [34]. Recent studies have identified antibiotic residues, antibiotic-resistant bacteria, and antibiotic resistance genes (ARGs) in various environments impacted by human activities. Overuse of antibiotics in fish farming exerts selective pressure on bacteria, favoring the development of antibiotic-resistant strains. Diverse bacteria, naturally resistant or acquiring resistance, have been detected in water samples from wastewater, recreational areas, surface waters, and drinking sources [35]. For instance, drinking water distribution systems, which include pipelines and water reservoirs, are susceptible to biofilm formation. Biofilms in these structures can harbor ARBs and ARGs, contributing to drinking water contamination. One study observed, through high-throughput quantitative polymerase chain reaction (PCR), about 285 ARGs, especially *sul1*, *ermB*, *tetQ*, *tetW*, *cfr*, *cmlA*, *fexA*, *fexB*, *floR*, and *qnrS*, as well as mobile genetic elements (MGEs), in water samples from drinking water treatment plants located at Hangzhou City, eastern China [36]. Another study detected MGEs, such as transposases and *intI-1* genes, suggesting their critical role in antibiotic resistance dissemination in drinking water. Sediment samples from water reservoirs located in central China have revealed the presence of 174 ARGs, with multidrug, sulfonamide, and vancomycin ARGs being the most prevalent. MGEs are identified as the main biotic factors contributing to ARG dissemination in sediment [37]. Similarly, most borehole and tap water samples in Ghana show no *E. coli* counts, and over 50% show no detectable *P. aeruginosa*. However, over half of the *E. coli* isolates were multidrug-resistant (MDR). *E. coli* isolates have shown high resistance to cefuroxime, trimethoprim–sulphamethoxazole, and amoxicillin–clavulanate [38]. *E. coli* from French drinking water with the *blaCTX-M-1* gene in an IncI1/ST3 plasmid demonstrates the presence of resistant bacteria and a specific resistance mechanism in a real-world setting. Linking the *blaCTX-M-1* gene found in *E. coli* to *K. pneumoniae* outbreaks and its presence in animal *E. coli* highlights the potential for cross-species transmission and environmental reservoirs. Significant differences have been observed in the structure of bacterial communities and the profiles of ARGs and MGEs between summer and winter samples from the Douhe Reservoir, China. Six specific MGE subtypes were identified as crucial for the spread of ARGs in both water and sediment. The evolution of bacterial communities appears to be the primary driver of changes in ARGs. In addition, one study also mentioned that environmental factors, particularly temperature, nitrates, total dissolved nitrogen, and total phosphorus, showed significant correlations with variations in bacterial communities, ARGs, and MGEs [39].

On the other hand, our aquatic environments are also under siege by a torrent of pollutants: industrial and agricultural runoff, municipal wastewater, oil spills, plastic waste, toxic chemicals, excess nutrients, sedimentation, radioactive materials, and mercury (Hg) contamination. These contaminants wreak havoc on aquatic ecosystems, killing species, disrupting food webs, and leaving water quality in a perilous state [40]. They also pose risks to human health through waterborne diseases and contaminated food sources. The increase in aquatic environment pollution has become a concerning and unfortunately common trend [41]. Human activities, including industrialization, agriculture, urbanization, and improper waste disposal, have led to the widespread contamination of water bodies, making water pollution a persistent and pervasive issue [42]. The rise in aquatic environment pollution potentially contribute to an increase in ABR, posing a significant public health concern [43]. One study found evidence showing the long-term impact of Hg contamination on increases in the persistence of environmental ARGs, specifically for tetracycline, sulfonamides, and trimethoprim. Interestingly, agriculturally important bacterial groups like Nitrospirae did not recover in the contaminated soils, suggesting the complex interplay between metal chemistries, especially Hg, soil pH, ABR, and microbial communities [44]. The widespread use of plastics also leads to the presence of ARGs in water bodies. Microplastics provide a perfect vector for microbes to colonize and exchange ARGs through a process called horizontal gene transfer (HGT) [45]. A previous study mentioned that aquatic environments serve as reservoirs for diverse bacterial populations, creating ideal conditions for the exchange and transfer of genetic material containing resistance genes through selective pressure that occurs when antibiotics are present in the environment [46]. This exposure creates conditions where bacteria are forced to develop ABR or mechanisms to survive and multiply, while susceptible bacteria are eliminated. The continued overuse or presence of antibiotics sustains this selective pressure, allowing resistant strains to thrive and become dominant [47]. On the other hand, HGT is a crucial process, facilitating the rapid spread of resistance genes among bacteria. Through HGT mechanisms such as conjugation, transformation, and transduction, resistant bacteria can transfer genetic material containing resistance traits to other bacteria, even those of different species [48]. This exchange enables the swift dissemination of resistance genes throughout bacterial populations, contributing significantly to the widespread occurrence of antibiotic resistance. The interconnectedness of bacterial populations in aquatic environments facilitates this gene exchange, making it vital to monitor and understand these ecosystems to effectively combat ABR [49]. 

The occurrence of ABR in aquatic environments poses a significant threat not only to human health but also to a wide range of ecological and environmental aspects, including ecosystems and food webs, disrupting their delicate balance and resilience. When the delicate balance of microbial communities is thrown off, both nutrient cycling and ecosystem stability suffer. This disruption, often fueled by the rise of antibiotic-resistant bacteria, directly impacts the organisms within these systems, posing a major threat to their well-being [50]. This disruption can extend to soil ecosystems, affecting the intricate relationships between microorganisms and plants, thus influencing agricultural productivity and soil fertility. In agricultural settings, for instance, the use of antibiotics in livestock can foster the development of resistant strains that affect animal health, potentially impacting humans through the food chain [51]. Similarly, in aquatic environments, the presence of antibiotic-resistant bacteria can disrupt the health of aquatic organisms, leading to imbalances in the ecosystem. The impact on fisheries and aquaculture due to antibiotic-resistant bacteria can affect food production and access to protein sources for humans [52].

## 3. Potential Negative Effects of ABR in Aquatic Environments on Human and Animal Health

The increase in ABR in aquatic environments brings forth significant concerns for both human and animal health. It has a multifaceted impact on public health, food security, economies, food systems, and livelihoods. Antibiotic-resistant organisms, particularly bacteria, can infiltrate aquatic environments through a multitude of pathways, creating concerns for both the environment and public health [53]. Once in the water, these antibiotic residues can affect the aquatic microbiome, which includes the diverse communities of microorganisms in the water [54]. Alterations to the environmental microbiome can have cascading effects on ecosystem health, potentially disrupting the balance of microorganisms and other organisms within the aquatic ecosystem. This can lead to the transmission of resistant bacteria to humans and animals through various pathways, such as contaminated water sources and the consumption of seafood [55]. The potential for infections that are challenging to treat with conventional antibiotics poses a serious threat to public health. Moreover, the interconnected nature of the food chain means that bacteria in aquatic organisms may be transferred to humans, further increasing the risk of multidrug-resistant infections. The persistence of antibiotic residues and resistant bacteria in aquatic environments not only threatens the effectiveness of medical treatments but also raises concerns about their long-term impact on ecosystem health and the potential emergence of novel resistance mechanisms [56]. ABR from animals can reach humans through various modes of transmission, including the food chain, handling, processing, transport, storage, and the preparation of food products. The transmission can begin at the farm level and spread within and between communities. Inadequate antibiotic use is linked to the decreased ability of fish species to metabolize drugs effectively, leading to prolonged antibiotic residues in fish meat [57]. These residues can persist in the terrestrial ecosystem through the food chain, and an estimated 70–80% of active compounds are eliminated through feces, contributing to antibiotic dispersion in wastewater and influencing diverse ecosystems. The existence of ARGs in aquaculture environments increases the risk of human and animal exposure to bacteria carrying these resistance traits. If these resistant bacteria are transmitted to humans through the consumption of contaminated aquaculture products or direct contact, it can lead to infections that are challenging to treat with common antibiotics [58]. 

Aquaculture settings where antimicrobials are used may act as reservoirs for antimicrobial resistance genes. This means that bacteria within the aquaculture environment exposed to antimicrobials may develop resistance to these drugs [59]. The presence of antimicrobial resistance genes in aquaculture settings raises concerns about the potential transmission of resistant bacteria to humans and animals, posing a threat to public health. In addition, the occurrence of ABR in aquatic environments has also led to changes in the aquatic environment’s microbiome, which can have broader implications for ecosystem health and function. The microbiome plays a crucial role in nutrient cycling, biodiversity maintenance, carbon sequestration, and freshwater availability. Disruptions to these ecosystem functions can have cascading effects on the overall health and sustainability of aquatic ecosystems [60]. ABR also poses a significant threat to veterinary medicine, jeopardizing animal production and, consequently, food security. The World Bank projects a potential 11 percent decline in livestock production in low-income countries by 2050 due to the challenges presented by ABR. This projection suggests a substantial loss of nearly 4 percent of the world’s annual gross domestic product (GDP) by 2050 due to AMR [61]. Despite efforts, the levels and patterns of antimicrobial use in aquaculture globally are largely undocumented, limiting the development of targeted interventions and policies for antimicrobial stewardship. 

On the other hand, since the advent of penicillin in the mid-20th century, antimicrobial treatments have played a crucial role not only in human medicine but also in veterinary care. Apart from therapeutic and prophylactic uses, low doses of antimicrobials have been added to animal feed to promote faster growth. While an increasing number of countries prohibit the use of antimicrobials as growth promoters, it remains a common practice in many parts of the world. However, the growth-promoting use of antibiotics among animals is associated with drawbacks. It does not result in the irreversible destruction of harmful bacteria, and sublethal doses act as selective pressure, promoting the evolution of bacterial strains resistant to antibiotics. This poses threats not only to human health but also to animal health, welfare, and sustainable livestock production, with implications for food security and people’s livelihoods [62]. Antibiotics, including macrolides and sulfonamides, have been found to negatively affect the growth, development, and reproduction of aquatic organisms such as algae. Antibiotics can also damage the photosystems of plant cells and interfere with carbon dioxide transformation [63].

## 4. Types of ARGs in Aquatic Environment

The dissemination of ARGs is indeed a global concern, influenced by the widespread use of antibiotics in medical care and animal husbandry. Different categories of ARGs confer resistance to specific classes of antibiotics, including tetracyclines (tet), sulfonamides (sul), β-lactams (bla), macrolides (erm), aminoglycosides (aac), fluoroquinolones (fca), colistin (mcr), vancomycin (van), and multidrug resistance (MDR). Several factors contribute to the spread of ARGs in various environments. Intracellular ARGs (iARGs) are prominent in nutrient-rich environments, while extracellular ARGs (eARGs) are prevalent in aquatic environments [64]. eARGs can be adsorbed by soil and sediment particles, avoiding DNase degradation and persisting longer than iARGs. This highlights the critical role of eARGs in the environmental dissemination of antibiotic resistance. A global analysis of the resistome has identified common ARGs in different settings. Hospitals commonly exhibit multidrug, glycopeptide, and β-lactam ARGs (e.g., *mecA*, *vanA*, *vanB*, and *bla*) [65]. In farms, wastewater treatment plants (WWTPs), water, and soil, sulfonamide and tetracycline ARGs (*sul* and *tet*) are prevalent. The top 10 ARGs reported from Asia include *blaNDM-1, blaCTX-M-15*, *mecA*, *blaTEM-1*, *sul1*, *vanA*, *blaKPC-2*, *sul2*, *blaCTX-M-14*, and *blaOXA-48*, indicating resistance to β-lactam, multidrug, sulfonamide, and glycopeptide antibiotics [66].

The *blaCTX-M* gene encodes for beta-lactamase enzymes that can break down a wide range of beta-lactam antibiotics, including penicillin, cephalosporins, and carbapenems. Beta-lactam antibiotics are one of the most important classes of antibiotics used to treat bacterial infections. This gene is found in a variety of bacteria, including *Klebsiella pneumoniae*, *Escherichia coli*, and *Salmonella* spp. [67]. *blaCTX-M* genes can spread between bacteria through conjugation, transduction, and transformation [68]. One study mentioned that *blaTEM-1*, *blaCTX-M-15*, and *blaCMY-42* were found in *E. coli* from the Yamuna River in Delhi, India. It also revealed that these ARGs show plasmid-mediated HGT, which is indeed a coevolutionary process that plays a crucial role in the spread of ecologically important traits, including antibiotic resistance, virulence factors, and metabolic capabilities, among bacteria [69]. Similarly, a study conducted on the Lis River in central Portugal found that the low water quality across various sites was probably due to the continuous discharge of effluents along its path and the persistence of ongoing pollution inputs; the study that observed, out of 147 cefotaxime-resistant Enterobacteriaceae isolates, 46% of them carried *blaCTX-M*. The most common *blaCTX-M* variant is *blaCTX-M-15*. Sites with poorer water quality exhibit higher resistance rates and blaCTX-M prevalence, suggesting potential risks to human health associated with river water contamination [70]. Likewise, in a study conducted along three sites on the Tigris River, Iraq, 40 out of 67 bacterial isolates were identified as *E. coli* using the Vitek2 diagnostic method. The antibiotic sensitivity of these *E. coli* isolates was noted, focusing on seven antibiotics from the β-lactam and carbapenem classes. The results revealed considerable resistance, with *E. coli* showing high resistance rates to β-lactam antibiotics such as amoxicillin–clavulanic acid (AMC) (82.5%) and piperacillin (PRL) (62.5%). Additionally, the study investigated the presence of carbapenem resistance genes and identified two isolates (5%) with the *blaVIM* gene and one isolate (2.5%) with the *blaNDM* gene, emphasizing the potential for carbapenem resistance in the *E. coli* population [71]. 

Tetracycline resistance genes (tet) and the associated efflux protein genes have been detected in environmental samples. A pharmaceutical facility (PFI) showed the highest occurrence of tetracycline resistance genes. The efflux gene *tet(G)* is the most prevalent among the *tet* genes found in all metagenomic deoxyribonucleic acid (DNA) samples. *TetA*, *tetK*, *tetC*, *tetE*, and *tetM* are genes associated with tetracycline resistance in bacteria [72]. These genes code for efflux pumps and ribosomal protection proteins, which enable bacteria to resist the effects of tetracycline antibiotics, which are used to treat a wide range of bacterial infections, including pneumonia, acne, and Lyme disease [73]. The presence of *tet* genes in environmental bacteria can contribute to the development of ABR. Tetracyclines are widely used in both human and veterinary medicine, and when these antibiotics are released into the environment through wastewater or agricultural runoff, they can select for bacteria-carrying *tet* genes. This can lead to increased ABR in the environment [74]. To support this statement, a study conducted on pig slaughterhouses based in Indonesia identified various tetracycline resistance genes in *E. coli* from floor surfaces and effluent samples from pig slaughterhouses [75]. Genes like *tetA, tetC, tetM, tetO, tetX*, and *tetE* were detected. *tetO* was dominant on floors (60%), while tetA dominated effluents (50%). The *tetA* and *tetO* combination was common (15%). This highlights the transmission of resistance genes from pigs to the environment, posing a significant public health threat. Another study found that several ARGs were widely present, especially *tetA* and *tetM*, with the highest detection rates in samples derived from a wastewater treatment plant in Guangzhou [76].

The *floR* gene confers resistance to florfenicol, which is a broad-spectrum antibiotic used in veterinary medicine, particularly in the treatment of bacterial infections in animals [77]. The presence of the *floR* gene in bacteria can lead to resistance against florfenicol, limiting the effectiveness of this antibiotic in treating infections caused by such bacteria. The *floR* gene encodes a protein that pumps florfenicol out of the bacterial cell. This prevents florfenicol from accumulating inside the cell and kills the bacterium. Bacteria that carry the *floR* gene are often resistant to other antibiotics as well, making them difficult to treat [78]. In one study, 296 bacterial isolates obtained from drinking water distribution systems in southwestern Nigeria were screened for drug resistance, and 30 isolates, including Pseudomonas, Serratia, Proteus, Acinetobacter, and *Providencia rettgeri*, were chosen based on their multidrug resistance and their resistance to the veterinary antibiotic florfenicol [79]. About 11 out of the 30 isolates were resistant to this particular antibiotic, indicating the widespread distribution of this resistance gene in the drinking water systems of Nigeria. The presence of *floR* in various bacterial genera without selective enrichment suggests that further research is needed to understand whether antibiotic use practices, both in humans and animals, contribute to the proliferation of this resistance gene and its potential impact on human and animal health [80]. 

The *sul/I* gene, or sulfonamide resistance gene I, is associated with resistance to sulfonamide antibiotics, which are a class of antimicrobial drugs [81]. Sulfonamides, also known as sulfa drugs, are synthetic antimicrobial agents that inhibit the growth of bacteria by interfering with the synthesis of folic acid, which is essential for bacterial growth. The *sul/I* gene encodes an enzyme called dihydropteroate synthase, which sulfonamides target. Bacteria that carry the *sul/I* gene are resistant to the inhibitory effects of sulfonamide antibiotics. This gene is often found in MGEs, such as plasmids, which can facilitate its transfer between different bacterial species, contributing to the spread of sulfonamide resistance in bacterial populations [82]. One study examined sulfonamide resistance genes (*sul* genes) in *E. coli* isolates from shrimp and pork in China. It found high prevalence rates of *sul1* and *sul2* in these isolates. The genes were located on plasmids and/or chromosomes and transferred through conjugation. Various replicon types were identified, with IncF being common among plasmids. Insertion sequences, especially IS26, were present in many sul gene-related fragments. Sul1 was frequently associated with class 1 integrons and other resistance genes, while sul3 had less diversity in its genetic environment. These findings suggest that horizontal gene transfer plays a significant role in sul gene transmission [83]. The Northern Yellow Sea, a densely populated and industrialized region crucial for drinking water and fishing, has exhibited high levels of sulfonamide antibiotics. These antibiotics have been linked to the presence of antibiotic-resistant bacteria, posing significant risks to human health, animal well-being, and the environment [84]. Another study mentioned the presence of multiple ARGs, such as *blaCTX, tetA, floR*, and *sul2*, in *Salmonella* spp. isolated from *Oreochromis niloticus* in Brazil, indicating the occurrence of ABR [85]. Another study showed high dissemination of extended-spectrum beta-lactamase (ESBL)-producing Gram-negative bacteria in Lake Água Preta, Brazil; nearly 88% of the isolates exhibited resistance to antibiotics from at least three different classes of antibiotics [86]. A significant proportion (84.7%) of the isolated strains derived in the lagoon of Bizerte in Tunisia exhibited a multi-resistant phenotype, indicating resistance to multiple classes of antibiotics [87]. Table 1 shows a summary of the presence of ARGs found in various aquatic environments. 

## 5. Bacterial Extracellular Vesicles (BEV)

Extracellular vesicles (EVs) have been around since the 1960s, when there were sporadic publications in scholarly journals. In contrast to mammalian EVs, bacterial EVs are more easily seen in routine electron microscopy, and numerous researchers have recorded these observations [88]. However, the challenge lies in understanding the biological meaning and functions of bacterial EVs. Because polar lipids have a tendency to form vesicular structures in aqueous solution, it was first thought that bacterial EVs were cellular debris or dust left over from the breakdown of dead cells, specifically, their lipid membranes [89]. However, evidence emerged that bacterial EV production requires metabolic activity, and structural and functional similarities between bacterial and mammalian EVs suggest that bacterial EVs are released by living bacteria [90]. In laboratory conditions, EVs have been demonstrated to mediate the delivery of a variety of molecules, both within and between species, such as through involvement in toxin delivery, biofilm formation, quorum sensing, defense against antimicrobials, nutrient acquisition, horizontal gene transfer, and ATP transfer. Studies conducted in the field have verified the existence of microbial EVs transporting a range of cargo in a variety of settings, including aquatic environments, spanning oceans, seas, lakes, rivers, and other water bodies; sewage; indoor dust; and seawater [91]. Previous research has also identified ARGs and MGEs in indoor dust EVs, emphasizing the need for further investigation into the origin, cargo, and functions of EVs in environmental microbiota [92]. Most studies on Gram-negative bacteria, both pathogenic and non-pathogenic, have demonstrated evidence of bacterial EV (BEV) production under diverse culture conditions and in various natural environments [93]. Notably, a study by Biller et al. (2014) highlighted the production of membrane vesicles (MVs) by strains from the dominant Prochlorococcus genus of marine cyanobacteria, suggesting that marine phototrophic bacteria release MVs both in situ and in vitro [94]. Scientists estimate that the global production of EVs by the cyanobacterium Prochlorococcus alone reaches a billion per day, adding a significant amount of carbon to the ocean’s nutrient pool [95]. The similar study suggested that marine MVs could be involved in phage defense, carbon cycling, horizontal gene transfer, and cellular communication. Nevertheless, nothing is known about the roles and existence of EVs in marine microorganisms [94]. 

Specifically, the classification of prokaryotic EVs divides them into two main categories: Gram-negative BEVs and Gram-positive BEVs. These distinctions encompass various subtypes, including outer-membrane vesicles (OMVs), outer–inner membrane vesicles (O-IMVs), cytoplasmic membrane vesicles (CMVs), and tube-shaped membranous structures (TSMSs) [96]. OMVs, originating from Gram-negative bacteria, possess a phospholipid inner leaflet and an outer leaflet composed of lipopolysaccharide (LPS), outer membrane proteins, and periplasmic proteins. Their formation involves the budding of the membrane, trapping various molecules such as LPS, lipoproteins, outer membrane proteins, and flagellin. These OMVs carry cargo like ribonucleic acid (RNA), DNA, proteins, and virulence factors, highlighting their potential role in bacterial communication and pathogenesis [97,98]. O-IMVs are also natural secretions of some Gram-negative bacteria, such as *Neisseria gonorrhoeae*, *Pseudomonas aeruginosa* (*P. aeruginosa*) PAO1, and *Acinetobacter baumannii* (*A. baumannii*) AB41. The percentage of O-IMVs in all vesicles varies between 0.23 and 1.2%; they feature a double-bilayer structure and specifically facilitate the transfer of local intracellular components such as DNA during their formation process [99]. CMVs are the other type of membrane vesicles (MVs) and are primarily associated with Gram-positive bacteria, including *Bacillus subtilis*, *Bacillus anthracis*, and *Staphylococcus aureus* (*S. aureus)*, but have also been observed in Gram-negative bacteria like *Acidiphilium cryptum* JF-5 under stress conditions [100]. This suggests the potential for CMV release from different bacterial groups in specific circumstances [101]. TSMSs, also known as nanotubes, nanowires, or nanopods, represent another type of BEVs. These structures are produced by both Gram-positive and Gram-negative bacteria, serving as bridges between cells and facilitating the exchange of components [102]. TSMSs are characterized by an average tube width of 50–70 nm and connect cells within biofilms at the periplasmic level, enabling social interactions between bacteria. *Myxococcus xanthus*, the production of OMV chains, and TSMSs interconnect cells, facilitating the transfer of molecules, including membrane proteins, between bacterial cells [92]. Understanding the diversity and roles of these different types of BEVs is crucial in comprehending their contributions to bacterial communication, biofilm formation, and intercellular interactions within microbial communities [103]. Understanding the functions of BEVs in these environments is crucial for deciphering microbial interactions, comprehending antibiotic resistance dissemination, and managing the ecological impact of microbial communities in aquatic habitats. 

## 6. Example of a Bacterial EV Responsible for Antibiotic Resistance

BEVs have been increasingly recognized for their role in mediating various functions, including ABR. The fight against ABR takes an unexpected turn in the vast realms of aquatic environments. BEVs, the tiny bubble-like messengers released by bacteria, play a surprising role in spreading resistance against critical antibiotics like β-lactams. These BEVs, acting like miniature Trojan Horses, can carry potent β-lactamase enzymes that are packaged into BEVs and released into the surrounding water, eventually causing an imbalance in the aquatic ecosystems [104].

In the case of *A. baumannii*, the release of oxacillinase (OXA)-58 via EVs presents a significant challenge in combating antibiotic resistance. OXA-58, a type of D β-lactamase, functions by hydrolyzing carbapenem antibiotics. This action enables the protection of carbapenem-susceptible bacteria from being killed by carbapenems, one of the last lines of defense against multidrug-resistant bacteria like *A. baumannii*. The dissemination of antibiotic resistance genes or enzymes through EVs between bacterial cells amplifies the potential for resistance transmission, making the treatment of infections caused by these bacteria more challenging [105]. In another example, OMVs derived from *Stenotrophomonas maltophilia* and other resistant bacterial strains, including *E. coli*, *Moraxella catarrhalis,* and *Bacteroides fragilis*, carry enzymes like β-lactamases [106]. The presence of these enzymes points to the protective role of vesicles when cells are in stress conditions. These enzymes not only shield the producing bacteria from β-lactam antibiotics but also confer resistance to other bacterial species, like P. aeruginosa and *Burkholderia cenocepacia* [107]. Meanwhile, an experimental study observed that the introduction of cationic poly(ionic liquid)-based antimicrobial materials to bacteria such as *E.coli*, *S. aureus*, and *Vibrio fischeri* (*V. fischeri*) has contributed to an unexpected increase in bacterial nanotube formation between bacterial cells. It has been postulated that this could enhance the intraspecies exchange of ARGs, even spreading ARGs from pathogens to environmental microbes like *V. fischeri* [108]. In addition, *K. pneumoniae* HCD1 utilizes OMVs as a sophisticated tool for survival and resistance, potentially spreading resistance within bacterial communities. It harbors genes for three beta-lactamases, including the carbapenemase KPC-2, placing it among highly resistant strains [109]. In another case, *Haemophilus influenzae* (*H. influenzae*), produces β-lactamase and packages it into vesicles. These vesicles, containing β-lactamase from *H. influenzae*, might be released by bacteria, possibly Group A Streptococcus, to protect themselves from the effects of amoxicillin [110]. By producing and releasing vesicles with β-lactamase, these bacteria can create an environment where the antibiotic (amoxicillin) is neutralized, allowing them to survive and propagate. 

Biofilm antibiotic resistance is a significant challenge in the field of antimicrobial therapy. Biofilms are complex communities of microorganisms, such as bacteria, embedded in a self-produced extracellular matrix [111]. This structure is enclosed within a self-produced matrix of extracellular polymeric substances (EPSs), which form a matrix consisting of water, microbial cells, ions, homo- and heteropolysaccharides, lipids, proteins, extracellular nucleic acids (DNA and RNA), and other molecules. This matrix provides structural support and protection for the microorganisms within. The resistance of biofilms to antibiotics is often much higher than that of planktonic (free-floating) bacteria. EVs in biofilm bacteria play intriguing roles in the formation, structure, and function of biofilms. EVs released by biofilm-forming bacteria contribute to the resilience, communication, and interaction within these communities. EVs act as carriers of various molecules like proteins, lipids, nucleic acids, and signaling molecules. They facilitate communication between cells within the biofilm, allowing bacteria to exchange genetic material, virulence factors, and other bioactive molecules [112]. This communication aids in coordinating biofilm development and adaptation to environmental changes. For example, EVs derived from *P. aeruginosa* have been reported to transport the signaling molecule Pseudomonas quinolone signal (PQS), specifically, 2-heptyl-3-hydroxy-4(1H)-quinolone, which plays a pivotal role in regulating virulence factors, biofilm formation, iron acquisition, cytotoxicity, and the biogenesis of OMVs [113]. In *S. aureus*, EVs act as carriers for crucial factors vital to bacterial survival and virulence. These vesicles transport a range of significant components including β-lactamases, superantigens, toxins, coagulases, and proteins linked to the bacterium’s ability to adhere to host cells. By packaging and transporting these factors, *S. aureus* EVs play a role in facilitating bacterial interactions with host cells, contributing to the bacterium’s pathogenicity and ability to cause infections [114]. Research has also found that EVs associated with the cytosolic pore-forming toxins of *Streptococcus pneumoniae* bind to complement proteins, thereby promoting the pneumococcal evasion of complement-mediated opsonophagocytosis [115]. Likewise, *Bacillus anthracis*-derived EVs harbor biologically active components of the anthrax toxin. These vesicles contain elements of the anthrax toxin complex, which exhibit toxicity toward macrophages, a type of immune cell. Interestingly, exposure to these EVs can also induce a protective response in the host, potentially triggering immune mechanisms that aim to defend against anthrax infection. This dual nature of *B. anthracis* EVs, carrying both toxic components and triggering protective responses, underscores their complex role in host–pathogen interactions. Additionally, they can carry stress response proteins and molecules, aiding the biofilm in adapting to harsh conditions [116]. Table 2 shows a summary of the presence of BEVs responsible for antibiotic resistance. 

## 7. Potential Strategies to Address ABR in Aquatic Environments Using EVs

The increasing recognition of EVs in clinical applications has positioned them as valuable diagnostic tools for complex diseases and potential carriers for therapeutic delivery [117]. EVs, which are small, membrane-bound structures released by cells, have become the subject of intense research because of their role in intercellular communication across various biological systems. Two main areas of exploration emerge from their widespread distribution: the development of targeted antimicrobial delivery systems and insights into host–pathogen interactions during infections [118]. EVs derived from both plant and human hosts have shown promise in delivering natural antimicrobial cargo to combat invading fungal and bacterial pathogens. 

The concept of using biomarkers and exosomes in the context of ABR is an intriguing area of research. Biomarkers are measurable indicators of biological processes, conditions, or responses to treatment. They can be molecules, genes, or characteristics that are associated with a particular disease state or physiological condition [119]. Exosome cargo can be used as biomarkers, whereby the content of exosomes, including specific proteins or nucleic acids, could potentially serve as biomarkers for ABR. Analyzing the exosome cargo might provide insights into the resistance status of bacterial populations and could aid in the early identification and management of resistant infections [120].

This avenue of research holds the potential for designing more effective and targeted antimicrobial delivery systems. EVs can be engineered to carry natural antimicrobial cargo. Meanwhile, loading EVs with antimicrobial agents could enhance their targeted delivery to combat invading fungal and bacterial pathogens. The enzymatic degradation of antibiotics in water environments can be accomplished by using exosomes loaded with enzymes like β-lactamases and tetracycline hydrolases to degrade different antibiotics in water samples [121].

The modification of host EV populations is being explored to enhance their pathogen-killing capabilities, laying the foundation for advanced therapeutic options against challenging-to-treat pathogens. This is possible since EVs can be functionalized or engineered to enhance their drug delivery capabilities. Surface modification can improve targeting specificity and increase therapeutic efficacy. The incorporation of specific ligands or surface proteins on EVs allows for targeted delivery to particular cell types. This may serve as a foundation for the development of advanced therapeutic options, particularly against challenging-to-treat pathogens [122]. 

In addition, we can introduce BEVs derived from beneficial bacteria that produce prebiotics or probiotics (live beneficial bacteria) into aquatic environments. Probiotic-derived EVs represent a novel avenue of research with potential therapeutic applications. These probiotics can inhibit pathogens through the production of antimicrobial agents, competitive exclusion, and other mechanisms. For example, bacteriocins, peptides with antimicrobial activity produced by probiotics, have been identified in EVs. These EVs may deliver bacteriocins to kill other bacteria, providing protection. They may contribute to the prevention and treatment of infectious diseases and modulate host immune responses [123]. BEVs have demonstrated potential in drug delivery. They enhance drug uptake and protect cargo from degradation, delivering bioactive substances in functional conditions to target cells. The loading of bioactive substances can occur in vivo during EV biogenesis or in vitro through techniques such as electroporation. This approach has been used to load EVs with compounds like gentamicin, siRNA, or gold nanoparticles [124]. OMVs from *P. aeruginosa*, both natural (n-OMVs) and gentamycin-induced (g-OMVs), have been shown to include a periplasmic 26-kDa autolysin according to research by Kadurugamuwa and Beveridge. It was discovered that autolysins, which are endogenous enzymes that hydrolyze peptidoglycan connections, lyse cells and break down peptidoglycan. Interestingly, g-OMVs are superior to n-OMVs and free antibiotics in their ability to lyse gentamicin-resistant *P. aeruginosa* cultures, pointing to a potential breakthrough in the fight against resistant bacteria. These “predatory” OMVs could represent a conceptual advance in the creation of antibiotics by demonstrating bacteriolytic activity against both Gram-positive and Gram-negative infections [125] Studies have also demonstrated that plants can absorb and accumulate various environmental contaminants, including diverse antibiotic molecules, from water and soil through their roots, stems, and leaves. Specific mechanisms involved in this process include adsorption, absorption, and translocation within the plant body. For instance, several plants like *Phragmites australis* and *Iris tectorum Maxim* have shown excellent removal rates for various antibiotics (fluoroquinolones and tetracyclines), exceeding 90% in some cases. Plant EVs could potentially play a role in engineering plant EVs to specifically bind and internalize antibiotic molecules, which could improve their removal efficiency. EVs loaded with specific enzymes or bacteria known to degrade certain antibiotic types could be delivered to plant roots, enabling targeted decontamination. Plant EVs could interact with beneficial microbes in the rhizosphere, promoting their antibiotic-degrading activity [126,127]. A summary of the potential use of BEVs in combating ABR is included in Table 3.

Recent advances in active incorporation techniques, including electroporation and sonication, have successfully enhanced the integration of drugs and therapeutic agents into BEVs. While these methods have not been explicitly employed for loading antibiotics into BEVs, their success in incorporating various substances underscores the potential of BEVs as antimicrobial agents through this versatile approach [128,129,130]. Figure 2 unveils ABR in aquatic environments, focusing on the role of BEVs and potential mitigation strategies. However, despite these promising applications, several challenges need to be addressed before bacterial EVs can be safely employed. These include the potential presence of virulence/cytotoxic factors in BEVs, difficulty in standardizing EV composition in each batch, and the necessity of targeting BEVs to specific tissues. Gram-positive BEVs, lacking LPS and generally being less toxic, may present a safer option for vaccine development compared with Gram-negative BEVs. 

## 8. Future Perspectives

As shown in this review, addressing ABR in aquatic environments will necessitate a comprehensive and proactive approach. Firstly, scientific research must delve deeper into understanding the intricate mechanisms through which BEVs contribute to the dissemination of resistance genes and the proliferation of multidrug-resistant bacteria in aquatic ecosystems. This research should focus on elucidating the dynamics of BEVs, their interactions with other environmental factors, and their role in facilitating horizontal gene transfer.

Moreover, innovative strategies need to be developed to harness the potential of BEVs in combating ABR. One avenue of exploration is the design of targeted antimicrobial delivery systems utilizing BEVs as carriers. These systems could be engineered to specifically target and eliminate antibiotic-resistant pathogens, thus mitigating the spread of resistance. Additionally, exploring the use of BEVs derived from beneficial bacteria with probiotic properties holds promise for modulating microbial communities in aquatic environments and reducing the prevalence of ABR.

Collaboration among various stakeholders will be crucial for implementing effective solutions. This includes cooperation between researchers, policymakers, industry representatives, and environmental organizations. Together, they can advocate for stringent regulations on antibiotic use in aquaculture and other industries, promote the adoption of sustainable practices, and improve wastewater treatment processes to minimize the antibiotic contamination of aquatic ecosystems.

Furthermore, public awareness and education campaigns are essential for engaging communities and fostering a collective understanding of the importance of preserving aquatic environments and addressing ABR. By raising awareness about the risks associated with ABR and the role of BEVs in its dissemination, individuals can be empowered to support initiatives aimed at protecting aquatic ecosystems and promoting responsible antibiotic use.

In summary, a future-oriented perspective on addressing ABR in aquatic environments involves advancing scientific knowledge, developing innovative strategies utilizing BEVs, fostering collaboration among stakeholders, implementing stringent regulations, and raising public awareness. By embracing these approaches, we can work toward effectively mitigating the spread of ABR and safeguarding the health of aquatic ecosystems and public well-being.

## 9. Summary and Conclusions

The aquatic environment is a vital ecosystem encompassing oceans, seas, lakes, and rivers, crucial for sustaining diverse species and providing essential ecosystem services. However, the rampant use of antibiotics in aquaculture and widespread human activities have led to the emergence of ABR within various antibiotic classes, including aminoglycosides, beta-lactams, (fluoro)quinolones, tetracycline, and sulfa groups. Understanding the diverse types of BEVs and their roles in aquatic environments is crucial to deciphering microbial interactions, comprehending antibiotic resistance dissemination, and managing the ecological impact of microbial communities. Moreover, the identification of specific ARGs in BEVs, such as blaCTX-M, tetA, floR, and sul/I, highlights the need for targeted strategies to mitigate the impact of ABR in aquatic ecosystems. Potential strategies to address antibiotic resistance in aquatic environments using BEVs involve leveraging their role in drug delivery. Modifying host EV populations to enhance pathogen-killing capabilities and introducing BEVs derived from beneficial bacteria with probiotic properties represent innovative approaches. However, challenges, including standardizing EV composition and addressing safety concerns, need to be overcome for practical applications (Figure 2). 

## Figures and Tables

**Figure 1 ijms-25-03080-f001:**
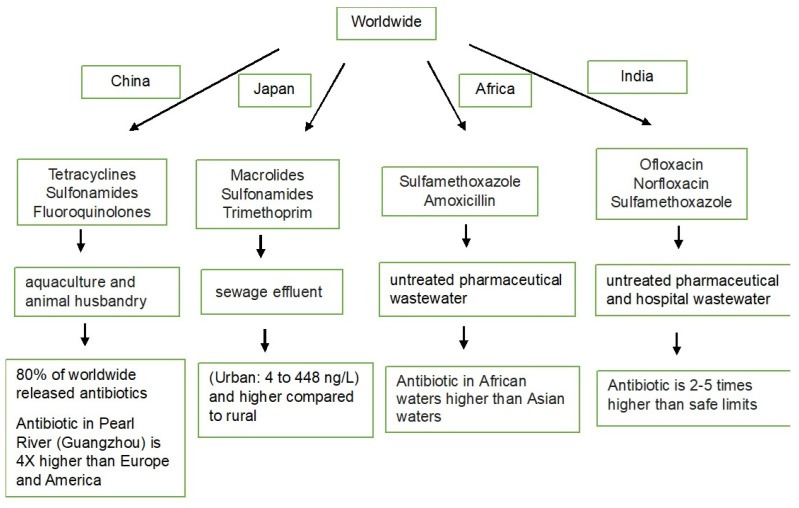
Antibiotic pollution in several countries.

**Figure 2 ijms-25-03080-f002:**
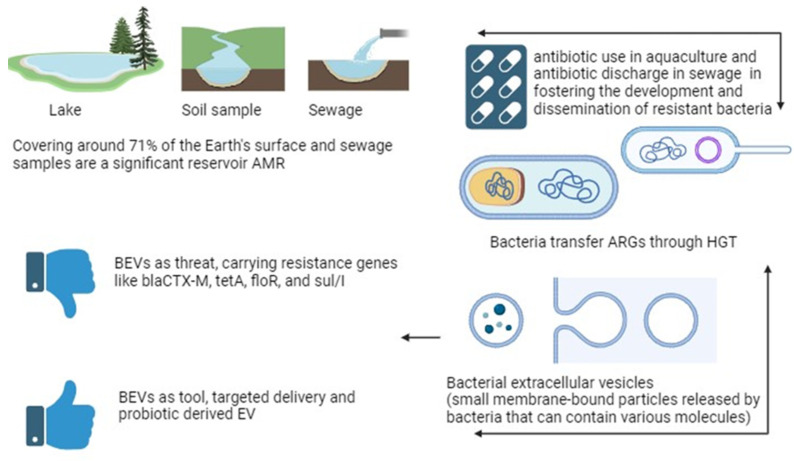
Unveiling antibiotic resistance (ABR) in aquatic environments focusing on the role of bacterial extracellular vesicles (BEVs) and potential mitigation strategies.

**Table 1 ijms-25-03080-t001:** Presence of various ARGs in aquatic environments.

ARGs	Antibiotic	Bacteria	Mechanism	Source	Citation
*blaCTX-M*	β-lactams (penicillins, cephalosporin, carbapenems)	*Klebsiella pneumoniae*, *Escherichia coli*, *Salmonella* spp.	Transfer through HGT (conjugation, transduction, transformation)	River, wastewater (United States)	[67,68,69,70]
*blaTEM-1, blaCTX-M-15, blaCMY-42*	β-lactams	*Escherichia coli*	Plasmid-mediated HGT	River (India)	[71]
*blaCTX-M*	β-lactams	Enterobacteriaceae	Plasmid-mediated HGT	River (Portugal)	[72]
*tetA, tetK, tetC, tetE, tetM*	Tetracyclines	*Escherichia coli*	Efflux pumps and ribosomal protection proteins	Pig slaughterhouses (Indonesia); wastewater, agricultural runoff (Guangzhou)	[76,77]
*floR*	Florfenicol	*Escherichia coli,* Pseudomonas, Serratia, Proteus, Acinetobacter, *Providencia rettgeri*	Florfenicol efflux pump	Drinking water (Southwestern Nigeria)	[80,81]
*sul/I*	Sulfonamides	*Escherichia coli*	Plasmid and/or chromosome transfer through conjugation	Sea and shrimp (China)	[82,83]
*blaCTX, tetA, floR, sul2*	Multiple	*Salmonella* spp., *E. coli*	Plasmid-mediated HGT	Fish (Kenya)	[84,85]
*blaCTX–M, blaTEM, blaSHV*	β-lactams	Extended-spectrum beta-lactamase Gram-negative bacteria	HGT	Lake (Brazil)	[86]
*blaCTX-M-1, blaOXA-1, blaTEM-1-a, qnrA, qnrB*	β-lactams and quinolone	Extended-spectrum beta-lactamase Gram-negative bacteria	Plasmid-mediated, efflux pump, and horizontal gene transfer	Lagoon (Tunisia)	[87]

**Table 2 ijms-25-03080-t002:** Examples of various bacterial EVs responsible for antibiotic resistance in aquatic environments.

Mechanism	Bacteria Involved	Resistance Genes/Enzymes	Citation
BEVs carrying β-lactamase enzymes	*Acinetobacter baumannii*, *Stenotrophomonas maltophilia*, *Escherichia coli*, *Moraxella catarrhalis*, *Bacteroides fragilis*	OXA-58 (carbapenemase), β-lactamases	[105]
Bacterial nanotube formation induced by cationic poly(ionic liquid)-based materials	*Escherichia coli*, *Staphylococcus aureus*, *Vibrio fischeri*	ARGs (not specified)	[108]
OMVs carrying β-lactamase enzymes	*Klebsiella pneumoniae* HCD1, *Haemophilus influenzae* (potentially packaged by Streptococcus)	KPC-2 (carbapenemase), β-lactamase	[106,107,108,109,110]
EVs contributing to biofilm formation and structure	*Pseudomonas aeruginosa*, *Staphylococcus aureus*, *Streptococcus pneumoniae*, *Bacillus anthracis*	PQS (signaling molecule), β-lactamases, superantigens, toxins, cytosolic pore-forming toxins, anthrax toxin components	[112,113,114,115,116]

**Table 3 ijms-25-03080-t003:** Application of BEVs to combat antibiotic resistance in aquatic environments.

Application	Description	Advantages	Challenges
BEVs as drug delivery vehicles	BEVs engineered to carry natural antimicrobial cargo or loaded with antimicrobial agents	Enhanced specificity and reduced side effects compared with conventional antibiotics	Lack of standardization
BEVs loaded with antibiotic-degrading enzymes or bacteria	Delivering BEVs loaded with enzymes or bacteria to plant roots for targeted antibiotic decontamination	Targeted approach to antibiotic removal	Ensuring BEV stability and efficient enzyme/bacteria activity
Plant EVs for antibiotic removal	Engineering plant EVs to specifically bind and internalize antibiotic molecules for improved removal efficiency	Environmentally friendly approach to antibiotic removal in water and soil	Understanding and manipulating plant EV–antibiotic interactions and ambiguous mechanisms
BEVs with probiotics	BEVs derived from beneficial bacteria that produce prebiotics or probiotics introduced into aquatic environments	Low immunogenicity and -probiotics can inhibit pathogens through various mechanisms	Ensuring BEV stability and viability in the environment
Modified host BEVs	Engineering EVs with specific ligands or surface proteins for targeted delivery to pathogens	Potential for developing new therapies against challenging-to-treat pathogens	Addressing potential safety concerns of modified EVs

## Data Availability

Not applicable.

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
