# Peer review of "Unseen Weapons: Bacterial Extracellular Vesicles and the Spread of Antibiotic Resistance in Aquatic Environments"

_ijms, 2024, doi:10.3390/ijms25063080_

Round 1
Reviewer 1 Report
Comments and Suggestions for Authors
The article focuses on important issues related to the resistance of microorganisms to antibiotics. The problem is quite common due to the widespread use of antibiotics, which are increasingly present in the aquatic environment. I suggest the authors provide some numerical data on the use of antibiotics (amounts in the world, divided by region) and their content in various water environments. It will enrich the article.
I would also suggest including a glossary of terms/abbreviations used in the article at the end of the article. This will facilitate broader understanding of the article, not only for specialists in the field of microbiology sensu stricte. I recommend enriching the article with tabular data, which will make it easier to compare the content contained in the article.
Author Response
Dear Reviewer 1,
Greetings!
We are pleased to submit our revised manuscript titled “Unseen Weapons: Bacterial Extracellular Vesicles and the Spread of Antibiotic Resistance in Aquatic Environment” for publication in your esteemed journal. We have highlighted the changes that was made in the manuscript according to the recommendation from the reviewer using point-by-point format.
Reply: Thank you for your recommendation.
- The article focuses on important issues related to the resistance of microorganisms to antibiotics. The problem is quite common due to the widespread use of antibiotics, which are increasingly present in the aquatic environment. I suggest the authors provide some numerical data on the use of antibiotics (amounts in the world, divided by region) and their content in various water environments. It will enrich the article.
Reply: Thank you for the suggestion. We have revised it accordingly.
- I would also suggest including a glossary of terms/abbreviations used in the article at the end of the article. This will facilitate broader understanding of the article, not only for specialists in the field of microbiology sensu stricte.
Reply: Thank you for your suggestion. We have added the abbreviations accordingly before the reference section.
- I recommend enriching the article with tabular data, which will make it easier to compare the content contained in the article.
Reply: Thank you for the suggestion. We have revised it accordingly.
Thank you for the opportunity to further explain our research hence we request you to consider the manuscript for publication in the esteemed journal and oblige.
Thank you.
Sincerely,
Barathan Muttiah

Reviewer 2 Report
Comments and Suggestions for Authors Authors have reviewed Bacterial Extracellular Vesicles and the Spread of Antibiotic Resistance in Aquatic Environment. In section 3. Types of ARGs in aquatic environment Authors have employed table to summary all of interesting parameters in a table 1. Please specifically indicate how did you select those reference which condition key words, year and from which resource have you qualified that those references have been collected. Please include reference numbers in each line in the table The graphs are recommended to illustrated the findings. Table 2 same comments as above. Figure 1 Please move figure 1 before the conclusion. Please enhance the size of illustration for bacterial extracellular vesicles. Please reduce the texts to less than 10 words in the figure.Author Response
Dear Reviewer 2,
Greetings!
We are pleased to submit our revised manuscript titled “Unseen Weapons: Bacterial Extracellular Vesicles and the Spread of Antibiotic Resistance in Aquatic Environment” for publication in your esteemed journal. We have highlighted the changes that was made in the manuscript according to the recommendation from the reviewer using point-by-point format.
Reply: Thank you for your recommendation.
- Types of ARGs in aquatic environment Authors have employed table to summary all of interesting parameters in a table 1. Please specifically indicate how did you select those reference which condition key words, year and from which resource have you qualified that those references have been collected
Reply: Thank you for your suggestion. We have searched through literature search engines using keywords such as aquatic, antibiotic, resistance, ARG, and prevalence. We did not specifically filter the search according to years.
- Please include reference numbers in each line in the table 1 and 2
Reply: Thank you for your suggestion. We have revised it accordingly.
- The graphs are recommended to illustrated the findings
Reply: Thank you for your suggestion. We have revised it accordingly and included table.
- Figure 1 Please move figure 1 before the conclusion. Please enhance the size of illustration for bacterial extracellular vesicles. Please reduce the texts to less than 10 words in the figure.
Reply: Thank you for your suggestion. We have revised it accordingly.
Thank you for the opportunity to further explain our research hence we request you to consider the manuscript for publication in the esteemed journal and oblige.
Thank you.
Sincerely,
Barathan Muttiah

Reviewer 3 Report
Comments and Suggestions for Authors
Dear Authors,
the manuscript you presented is interesting and valuable to broad range of readers. In my opinion it can be suitable for publication in IJMS, after some corrections and improvements.
My comments to help you improve the paper are below:
In terms of merit, this paper is of high quality. It provides comprehensive information about the BEVs and their potential application in fighting ARB. It contains two tables and a figure. I suggest that adding one more table, that summarizes the most important data related to EVs as carriers of substances would enhance the paper’s attractiveness. In fact, data summarized in tables makes it easier for the potential readers to find information they are looking for and therefore increase the potential citations of the paper.
The English language used in the manuscript is generally good. There are some sentences that would need rephrasing, due to grammatical incorrectness, but the paper is readable and understandable.
The title of the first chapter should be narrowed down. “Aquatic environment” is so broad that you can actually describe anything associated with aquatic environment, but this should not be the case in this review.
l. 54 – it should be subtherapeutic antibiotic doses
l. 62: Aeromonas and Vibrio are Latin names so please use italics (refers to all Latin names throughout the text).
l. 66-68: This statement would greatly benefit from providing some examples.
l. 69: Better use “On the other hand”
l. 69-73: this sentence is too long and grammatically incorrect.
l. 105-107: this sentence is also grammatically incorrect.
Section 3 should be started from e.g. mentioning the most frequent types of ARGs and then their characteristics.
l. 124-128 – how does this study refer to aquatic environment?
Table 1 needs references for the listed data.
l. 222: a comma is missing between water bodies and sewage
Table 2 needs references, too, for the data provided.
The chapter 7, i.e. Conclusions should be renamed to Summary and conclusions, as the majority of this section is the summary of information provided throughout the manuscript.
The figure at the end of the manuscript, summarizing the most important information is a brilliant idea. However, I suggest to shorten the text a bit, e.g. by ablative absolute or others. Then, the text would become more legible
Comments on the Quality of English Language
The English language used in the manuscript is generally good. There are some sentences that would need rephrasing, due to grammatical incorrectness, but the paper is readable and understandable.
I think that the paper does not need extensive English improvement. Simply read the manuscript one more time and I'm sure you'll find the problematic sentences, which will allow you to correct them.
Author Response
Dear Reviewer 3,
Greetings!
We are pleased to submit our revised manuscript titled “Unseen Weapons: Bacterial Extracellular Vesicles and the Spread of Antibiotic Resistance in Aquatic Environment” for publication in your esteemed journal. We have highlighted the changes that was made in the manuscript according to the recommendation from the reviewer using point-by-point format.
The manuscript you presented is interesting and valuable to broad range of readers. In my opinion it can be suitable for publication in IJMS, after some corrections and improvements.
Reply: Thank you for your recommendation.
- In terms of merit, this paper is of high quality. It provides comprehensive information about the BEVs and their potential application in fighting ARB. It contains two tables and a figure. I suggest that adding one more table, that summarizes the most important data related to EVs as carriers of substances would enhance the paper’s attractiveness. In fact, data summarized in tables makes it easier for the potential readers to find information they are looking for and therefore increase the potential citations of the paper.
Reply: Thank you for your suggestion. We have improved our manuscript accordimgly.
- The English language used in the manuscript is generally good. There are some sentences that would need rephrasing, due to grammatical incorrectness, but the paper is readable and understandable.
Reply: Thank you for the suggestion. Grammatical errors have been checked and corrected throughout the manuscript (various lines)
- The title of the first chapter should be narrowed down. “Aquatic environment” is so broad that you can actually describe anything associated with aquatic environment, but this should not be the case in this review.
Reply: Thank you for the suggestion. We have revised the subtitle accordingly.
- 54 – it should be subtherapeutic antibiotic doses
Reply: Thank you for pointing it out. We have changed it accordingly.
- 62: Aeromonas and Vibrio are Latin names so please use italics (refers to all Latin names throughout the text).
Reply: Thank you for pointing it out. We have changed them accordingly
- 66-68: This statement would greatly benefit from providing some examples.
Reply: Thank you for the suggestion. We have now added some suitable examples (line 103-123).
- 69: Better use “On the other hand”
Reply: Reply: Thank you for the suggestion. We have revised it accordingly.
- 69-73: this sentence is too long and grammatically incorrect.
Reply: Thank you for the suggestion. We have revised the paragraph accordingly (line 125-129).
- 105-107: this sentence is also grammatically incorrect.
Reply: Thank you for the suggestion. We have revised them accordingly (line 161-165).
- Section 3 should be started from e.g. mentioning the most frequent types of ARGs and then their characteristics.
Reply: Thank you for the suggestion. We have revised them accordingly (line 235-251).
- 124-128 – how does this study refer to aquatic environment?
Reply: Thank you for the suggestion. We have removed the sentence from the manuscript.
- Table 1 needs references for the listed data.
Reply: Thank you for the suggestion. We have added it accordingly.
- 222: a comma is missing between water bodies and sewage
Reply: Thank you for pointing it out. We have added it accordingly (line 365)
- Table 2 needs references, too, for the data provided.
Reply: Thank you for the suggestion. We have added it accordingly.
- The chapter 7, i.e. Conclusions should be renamed to Summary and conclusions, as the majority of this section is the summary of information provided throughout the manuscript.
Reply: Thank you for the suggestion. We have revised them accordingly
Thank you for the opportunity to further explain our research hence we request you to consider the manuscript for publication in the esteemed journal and oblige.
Thank you.
Sincerely,
Barathan Muttiah

Reviewer 4 Report
Comments and Suggestions for Authors
2. The names of the pathogens should be written in italics. Please correct.
3. Could the Authors add in the review a description of the possible negative effects on human and animal health of the increase of antibiotic resistance in aquatic environments?
Comments on the Quality of English LanguageI detected some English composition mistakes. A detailed copyediting of the manuscript would be helpful
Author Response
Dear Reviewer 4,
Greetings!
We are pleased to submit our revised manuscript titled “Unseen Weapons: Bacterial Extracellular Vesicles and the Spread of Antibiotic Resistance in Aquatic Environment” for publication in your esteemed journal. We have highlighted the changes that was made in the manuscript according to the recommendation from the reviewer using point-by-point format.
Reviewer 4
- The names of the pathogens should be written in italics. Please correct.
Reply: Thank you for the comment. We have corrected them accordingly (various lines).
- Could the Authors add in the review a description of the possible negative effects on human and animal health of the increase of antibiotic resistance in aquatic environments?
Reply: Thank you for the suggestion. We have added a section on the possible negative effects of ABR on human and animal health. (line 153-210).
- I detected some English composition mistakes. A detailed copyediting of the manuscript would be helpful
Reply: Thank you for pointing it out. Grammatical errors have been checked and corrected throughout the manuscript (various lines).
Thank you for the opportunity to further explain our research hence we request you to consider the manuscript for publication in the esteemed journal and oblige.
Thank you.
Sincerely,
Barathan Muttiah

Round 2
Reviewer 1 Report
Comments and Suggestions for Authors
Accept.
Author Response
Thank you so much
Reviewer 2 Report
Comments and Suggestions for Authors
Authors have improved the manuscript. However, there are still reference style and figure citations need to be improved.
Reference should be cited as [number] not (number).
Reference style should be MDPI style. Please following the guide https://www.mdpi.com/authors/references
Figure 1
Figure 1 is not cited in the text.
The font of text is very small.
Please reduce the words number in the figure.
Figure 2
Please cite shortly before it appears (not only at end of summary).
Table 2 is out of margin
Table 3
please remove the vertical lines in the table
Author Response
Dear Reviewer 2,
Greetings!
We are pleased to submit our revised manuscript titled “Unseen Weapons: Bacterial Extracellular Vesicles and the Spread of Antibiotic Resistance in Aquatic Environment” for publication in your esteemed journal. We have highlighted the changes that was made in the manuscript according to the recommendation from the reviewer using point-by-point format.
Reply: Thank you for your recommendation.
- Reference should be cited as [number] not (number). Reference style should be MDPI style. Please following the guide https://www.mdpi.com/authors/references
Reply: Thank you for your comment. We have revised it accordingly.
- Figure 1: Figure 1 is not cited in the text; the font of text is very small; please reduce the words number in the figure.
Reply: Thank you for your suggestion. We have revised it accordingly.
- Figure 2, Please cite shortly before it appears (not only at end of summary).
Reply: Thank you for your suggestion. We have revised it accordingly.
- Table 2 is out of the margin.
Reply: Thank you for your comment. We have revised it accordingly.
- Table 3, please remove the vertical lines in the table.
Reply: Thank you for your comment. We have revised it accordingly.
Thank you for the opportunity to further explain our research hence we request you to consider the manuscript for publication in the esteemed journal and oblige.
Thank you.
Sincerely,
Barathan Muttiah

Round 3
Reviewer 2 Report
Comments and Suggestions for Authors
THe reference style has been improved.
there are two times section 4 in line 335 and 448 respectively.
Please correct them.
Line 664-674
Please delete the gaps
Table 2
the title of table 2 should be on the top of table.
Please remove the vertical lines on the table.
Table 3
Table 3 was not mentioned in the text. please introduce it shortly it was appear.
Please remove the vertical lines on the table.
Reference
Please change the font of following references
line 721-722
Please change the font of 'Proc. Indian Natl. Sci. Acad. 721 2022,88,160–171. '
line 841
Clin Infect Dis.
Line 871
Jahantigh, M.; Samadi, K.; Dizaji, R. E.; Salari, S. Antimicrobial resistance and prevalence of tetracycline 870 resistance genes in Escherichia coli isolated from lesions of colibacillosis in broiler chickens in Sistan, Iran. BMC Vet. Res. 2020, 16, 267.
line 873
Cold Spring Harb. Perspect. Med. 2016,
Author Response
Dear Reviewer 2,
Greetings!
We are pleased to submit our revised manuscript titled “Unseen Weapons: Bacterial Extracellular Vesicles and the Spread of Antibiotic Resistance in Aquatic Environment” for publication in your esteemed journal. We have highlighted the changes that was made in the manuscript according to the recommendation from the reviewer using point-by-point format.
Reply: Thank you for your recommendation.
- The reference style has been improved.
Reply: Thank you for your recommendation.
- there are two times section 4 in line 335 and 448 respectively. Please correct them
Reply: Thank you for your suggestion. We have revised it accordingly.
- Line 664-674, Please delete the gaps
Reply: Thank you for your suggestion. We have revised it accordingly.
- Table 2, the title of table 2 should be on the top of table, Please remove the vertical lines on the table.
Reply: Thank you for your suggestion. We have revised it accordingly
- Table 3,Table 3 was not mentioned in the text. please introduce it shortly it was appear, Please remove the vertical lines on the table.
Reply: Thank you for your suggestion. We have revised it accordingly
- Reference, Please change the font of following references, line 721-722, Please change the font of 'Proc. Indian Natl. Sci. Acad. 721 2022,88,160–171. '
Reply: Thank you for your suggestion. We have revised it accordingly.
- line 841, Clin Infect Dis.
Reply: Thank you for your suggestion. We have revised it accordingly.
- Line 871, Jahantigh, M.; Samadi, K.; Dizaji, R. E.; Salari, S. Antimicrobial resistance and prevalence of tetracycline 870 resistance genes in Escherichia coli isolated from lesions of colibacillosis in broiler chickens in Sistan, Iran. BMC Vet. Res. 2020, 16, 267.
Reply: Thank you for your suggestion. We have revised it accordingly.
- line 873, Cold Spring Harb. Perspect. Med. 2016,
Reply: Thank you for your suggestion. We have revised it accordingly.
Thank you for the opportunity to further explain our research hence we request you to consider the manuscript for publication in the esteemed journal and oblige.
Thank you.
Sincerely,
Barathan Muttiah

Round 4
Reviewer 2 Report
Comments and Suggestions for Authors
Authors have improved references style.
Figures have not been mentioned and introduced in the manuscript before it appears.
Table 2 and table 3 are full size tables. Please move table title starting from 0 from left margin. Not from 4.6 cm.
Figure 1 and figure 2. Please move the frame of the figures.
Figure 2. Please move figure 2 starting from 4.6 cm from left side of margin.
Conclusion
Requests more future perspective.
Please check the MDPI style for authors contribution in the template.
Author Response
Dear Reviewer 2,
Greetings!
We are pleased to submit our revised manuscript titled “Unseen Weapons: Bacterial Extracellular Vesicles and the Spread of Antibiotic Resistance in Aquatic Environment” for publication in your esteemed journal. We have highlighted the changes that was made in the manuscript according to the recommendation from the reviewer using point-by-point format.
Reply: Thank you for your recommendation.
- Authors have improved references style.
Reply: Thank you for your recommendation.
- Figures have not been mentioned and introduced in the manuscript before it appears.
Reply: Thank you for your comments. Figure 1 has been mentioned at line 140 and Figure 2 has been mentioned at line 656.
- Table 2 and table 3 are full size tables. Please move table title starting from 0 from left margin. Not from 4.6 cm.
Reply: Thank you for your suggestion. We have revised it accordingly.
- Figure 1 and figure 2. Please move the frame of the figures.
Reply: Thank you for your suggestion. We have revised it accordingly.
- Figure 2. Please move figure 2 starting from 4.6 cm from left side of margin.
Reply: Thank you for your suggestion. We have revised it accordingly.
- Conclusion, Requests more future perspective.
Reply: Thank you for your suggestion. We have revised it accordingly. Line 673-703
- Please check the MDPI style for authors contribution in the template.
Reply: Thank you for your suggestion. We have revised it accordingly.
Thank you for the opportunity to further explain our research hence we request you to consider the manuscript for publication in the esteemed journal and oblige.
Thank you.
Sincerely,
Barathan Muttiah

Round 5
Reviewer 2 Report
Comments and Suggestions for Authors
Author has improved figure style.
Table 2 title
Table 2. should be bold, but ' Example of various bacterial EV responsible for antibiotic resistance in aquatic environment.' should be no bold.
Please change line 584 with single line so far there is a gap between title and table.
Table 3 title
only Table 3 is in bold, 'Application of BEVs to combat antibiotic resistance in aquatic environment' should be no bold.
Figure 1
Please use font 9 for the figure legend
figure 2
please remove a line on the top of image
Please use font 9 for the figure legend
Always use italics to write E. coli.
line 172 , 207, 538
should be K. pneumoniae
Please check style of Authors contribution below and revise them.
Conceptualization, X.X. and Y.Y.; methodology, X.X.; software, X.X.;
Author Response
pls seee the attachment

Round 6
Reviewer 2 Report
Comments and Suggestions for Authors
The manuscript have been improved. However there are still two questions were remained.
Question 3. Please change line 584 with single line so far there is a gap between title and table. Reply: Thank you for your suggestion. We have revised it accordingly.
There is still big gap between title and table. Please use single line space for line 584.
Question 6
figure 2, please remove a line on the top of image, Please use font 9 for the figure legend Reply: Thank you for your suggestion. We have revised it accordingly. However there is no line on the image.
Please see the blue arrow pointed to the extra line on the top of image in the attached document.
